# Characterization of the Teaching Profile within the Framework of Education 4.0

**María Soledad Ramírez-Montoya** [1], **María Isabel Loaiza-Aguirre** [2,*], **Alexandra Zúñiga-Ojeda** [3] **and May Portuguez-Castro** [4]

1 Education at Tecnológico de Monterrey, School of Humanities, Monterrey 64849, Mexico; solramirez@tec.mx
2 Innovation, Training and Teacher Evaluation, Universidad Técnica Particular de Loja, Loja 1101608, Ecuador
3 Educational Sciences Faculty, Universidad Técnica Particular de Loja, Loja 1101608, Ecuador; azuniga@utpl.edu.ec
4 Institute for the Future of Education, Tecnologico de Monterrey, Monterrey 64849, Mexico; mayportuguezc@gmail.com
* Correspondence: miloaiza@utpl.edu.ec; Tel.: +593-992795566

**Abstract:** The authors of the Education 4.0 concept postulated a flexible combination of digital literacy, critical thinking, and problem-solving in educational environments linked to real-world scenarios. Therefore, teachers have been challenged to develop new methods and resources to integrate into their planning in order to help students develop these desirable and necessary skills; hence, the question: What are the characteristics of a teacher to consider within the framework of Education 4.0? This study was conducted in a higher education institution in Ecuador, with the aim to identify the teaching profile required in new undergraduate programs within the framework of Education 4.0 in order to contribute to decision-making about teacher recruitment, professional training and evaluation, human talent management, and institutional policies interested in connecting competencies with the needs of society. Descriptive and exploratory approaches, where we applied quantitative and qualitative instruments (surveys) to 337 undergraduate students in education programs and 313 graduates, were used. We also included interviews with 20 experts in the educational field and five focus groups with 32 chancellors, school principals, university professors, and specialists in the educational area. The data were triangulated, and the results were organized into the categories of (a) processes as facilitators (b), soft skills, (c) human sense, and (d) the use of technologies. The results outlined the profile of a professor as a specialized professional with competencies for innovation, complex problem solving, entrepreneurship, collaboration, international perspective, leadership, and connection with the needs of society. This research study may be of value to administrators, educational and social entrepreneurs, trainers, and policy-makers interested in implementing innovative training programs and in supporting management and policy decisions.

**Keywords:** teaching profile; educational innovation; higher education; entrepreneurship; Ecuador; Education 4.0

## 1. Introduction

New roles in sectors such as industry, society, government, and education have emerged due to the dizzying pace of technological development in recent years. The teacher has had to become a change agent and transform their knowledge, skills, and competencies. Hernandez-de-Menendez et al. [1] analyzed these changes as necessary because of Generation Z students, who are seen as true digital natives. They are a hyper-cognitive generation with student profiles that are different from the previous ones, requiring new forms of teaching and learning from experienced teachers in emerging technologies. In turn, the mediation of technologies causes changes in teachers' roles; therefore, educators need to be open to continuous improvements in instructional practices [2]. There is a need to develop ways to equalize the positions between instructors and students, while professors

need to be aware and provide opportunities for students to demonstrate creativity in their work. The role of teaching changes internally when influenced by external changes.

Through all these changes, the theme of digital literacy is constant. Developing digital competencies will help with better time management, greater confidence, the development of socio-constructive attitudes [3], increased student productivity, time savings for instructors, and better test results, among other things [4]. Technology-based learning experiences promote students' creative design competencies thus, positively impacting students' educational performance and skills development [5]. Creative designing also requires the right infrastructure with innovative learning spaces, which are thought to respond to future changes and students' needs and learning methods [6]. In this framework, digital literacy competencies linked to teacher training are outlined to improve educational quality.

In academic functioning, several stakeholders must be described to achieve the expected results: chancellors, managers, administrators, infrastructure professionals (networks, buildings, and facilities), and professors. Each of these agents contributes to enhance the functioning of the system. Recognizing this importance without attempting to diminish the contributions of each one, this study focused on the role of teachers in educational processes, recognizing them as important agents within the system and highlighting that this role has changed throughout history—going from Education 1.0 with mechanical processes to Education 2.0 with mass production, Education 3.0 where internet access brought a boost to training, and the present day of Education 4.0 with advanced connectivity that supports virtualization processes [7].

### 1.1. Education 4.0 in the Field of Teaching Processes

The transition from Education 1.0 to 4.0 has been associated with technological advances and the development of industry. In Education 1.0, around the industrial revolution of the 18th century, the teacher was the center of education because they were in charge of determining and disseminating the essential information that students needed. Processes were supported by educational technologies such as the typewriter, the mechanical printing press, the graphite pencil, and the ballpoint pen. In Education 2.0, the evolution was associated with the second industrial revolution of the early 20th century when the teacher began to change their role towards one of reference and processes were supported by the first electronic devices used in education such as printers, calculators, and computers. Education 3.0, within the framework of the third industrial revolution at the end of the 20th century, brought with it processes of computerization, automation, and control, where teachers began to work as collaborators, and processes were supported by resources such as multimedia, virtual tools, and laboratories. In Education 4.0, coming within the fourth industrial revolution, processes are accompanied by innovative pedagogical technologies and procedures, and the role of the teacher is that of mentor, reference, and collaborator in connection with digital transformations and virtualization processes [7].

The well-known four industrial revolution periods were mobilized with the progress of technology and the associated changes in education. Miranda et al. [8] defined Education 4.0 as the current period in which higher education institutions apply new learning methods, innovative didactic and management tools, and intelligent and sustainable infrastructures complemented by emerging technologies that improve the processes of knowledge generation and information transfer. Additionally, they proposed four central components of Education 4.0 to be used as references for the design of new educational innovation projects: (i) competencies, (ii) learning methods, (iii) information and communication technologies, and (iv) infrastructure. An Education 4.0 teacher must have the same competencies demanded of students: digital literacy, critical thinking, and problem-solving, a fact that is of great interest to employers, who are the main stakeholders in education [9].

In recent years, the topic of Education 4.0 has been of interest to work in different countries and regions. Figure 1 shows a map created the authors regarding the topic of Education 4.0 linked to teaching processes. In a search conducted in the SciVal system on 15 March 2021, with the themes of Education 4.0 and teaching, Malaysia, Mexico, Indonesia,

and Germany stood out as the countries with the most active authors on the subject [10]. Authors from these countries have worked on their Education 4.0 writings within the framework of higher education with the use of emerging technologies (such as virtual and augmented reality) and open innovation that are linked to industry and other social sectors. This visualization helps to detect the international and interregional context where the topic is being researched and to suggest that the analyzed problems of Education 4.0 and teaching are not unique and have the potential for other users to benefit from advances on the topic.

| | Countries/Regions | Scholarly Output ↓ | Views Count ⌄ | Field-Weighted Citation Impact ⌄ | Citation Count ⌄ |
|---|---|---|---|---|---|
| 1. | Malaysia | 26 | 1,049 | 1.31 | 33 |
| 2. | Indonesia | 8 | 310 | 0.24 | 2 |
| 3. | Mexico | 8 | 374 | 1.84 | 11 |
| 4. | Germany | 5 | 262 | 4.10 | 49 |
| 5. | Brazil | 4 | 46 | 0.00 | 0 |
| 6. | Czech Republic | 3 | 235 | 0.28 | 1 |
| 7. | Greece | 3 | 425 | 16.69 | 85 |
| 8. | Philippines | 3 | 43 | 0.87 | 2 |
| 9. | Russian Federation | 3 | 58 | 0.87 | 2 |
| 10. | Thailand | 3 | 158 | 2.59 | 20 |

**Figure 1.** Top 10 countries/regions in this research area, ranked by scholarly output [10].

Technologies bring teachers together to generate active learning spaces in the formative environments of Education 4.0. Education 4.0 is based on learning by doing, in which students are encouraged to learn and discover in unique ways through experimentation [11]. Prieto et al. [12] outlined an Industry 4.0 Technologies Laboratory (I4Tech Lab), a technological environment for academic research and industrial promotion of related technologies, supported by an active-learning teaching methodology. Another possibility is to use augmented reality in a modular learning system with an interactive virtual model of the equipment, technical data, and information processed in real-time [13]. Likewise, Caluza [14] stated that Education 4.0 encourages using advanced technologies to facilitate educational ecosystems, in which teachers must be proficient in information and communication technologies. In the same vein, the dynamic nature of the higher education ecosystem and the connectivity between the elements of Education 4.0 have been highlighted as knowledge, industry, and humanity [15].

Various research works agree that the teacher is the single most crucial factor that determines the quality and effectiveness of education. Serdyukov [16] stated that it is necessary to improve teacher training and lifelong learning. To achieve this goal, teachers must have good attitudes. The author additionally pointed out the importance of developing their teaching styles, motivation, skills, competencies, self-evaluation, self-efficacy, creativity, responsibility, autonomy to teach, ability to innovate, freedom from administrative pressures, working conditions, and supportive policies to have good teaching environments. Teaching roles change significantly when the environments are mediated by technology, so educators need to be open to continuous improvements in instructional practices.

*1.2. Linking the Teacher's Role in the Framework of Education 4.0 and Educational Innovation*

How is the teacher's role linked to the possibilities of educational innovation? One scenario involves the implementation of instructional strategies to create innovation in

learning. Seechaliao [17] determined that a teacher should use questioning, classroom discussion, self-directed study, inductive and deductive thinking, and social media and networks that engage students in learning activities. Similarly, in modern educational systems supported by technologies, educators must take the following factors seriously: the role of learners, new ways of constructing knowledge, the real possibility of continuous assessment, and direct and interactive communication with the community [18]. Other authors [19] concluded that support should be provided for formal and informal education so that frequent opportunities for joint planning and teaching can occur. Creating innovation requires a teacher to bring together three fundamental aspects: a brilliant idea, a macro-environment comprising the educational environment and society, and a micro-environment that enables innovation [16].

In particular, teachers should plan the learning activities to take place over an extended period so that students have time to develop and refine their work. Emphasis needs to be placed on providing a high-quality information and communication technology (ICT) infrastructure and ongoing support; its absence would be a potentially significant barrier to innovative teaching practices. This is the case of hybrid learning, which is defined as a combination of different instructional methods, pedagogical approaches, and technologies to enhance teaching and learning [20]. Hybrid-synchronous-learning designs result in more active learning than traditional classes [21]; however, the main problems faced in facilitating these types of lessons are related to communication and cognitive overload caused by dividing attention. An "intelligent learning environment" allows a teacher to adapt to different educational environments through knowledge, task support, learner sensitivity, context-sensitivity, reflection, and feedback [22]. The concept of an environment encompasses its design and development, how the learner is engaged, and to what extent the environment is effective and efficient. Its success is measured by the freedom of activities and the stimulation of ideas and results from the teacher's attitude to promote innovative ideas [23].

How prepared are teachers for Education 4.0? Goh and Abdul-Wahab [24] argued that current faculty professors may lack the pedagogy needed to teach in this "digitalized" world and are not competent enough to guide students into the new era of technology-driven experiences. They asserted that institutions need to move away from the traditional way of delivering knowledge and conducting research in order to adopt new ways that provide autonomy to educators to define goals in the formative process where students learn through technology [25], and the approach is learner-centered [26,27]. The teacher competencies required to perform in these environments must be identified [28]. Teaching and learning processes, innovation, and value-added experiences for students using technology are part of the concept of Higher Education 4.0. However, when searching the literature on the characterization of the teaching role in the context of Education 4.0, we found a notable absence. In a broader search on Education 4.0 and teaching, 87 publications appeared, and one of them [29] qualitatively described an Education 4.0 teacher as having technological skills, guidance skills, lifelong learning skills, and personal characteristics. A search was conducted in the SciVal system on 15 March 2021 and publications to see where Education 4.0 works have been focused; Figure 2 highlights the keywords on the terms of "Education 4.0" and "teaching," where augmented reality, Industry 4.0, education, learning, and industrial revolution were found to be topics that have been addressed in the publications. The figure also shows the topics that have been addressed to a lesser extent (digital devices, society, and multimodality) and could be topics to which educational research can contribute. Additionally, Education 4.0 was found to have been worked on in the following disciplines: computer science, education, engineering and industry, opening opportunities in the areas of health, arts, citizenship, data science, information, and communication technologies.

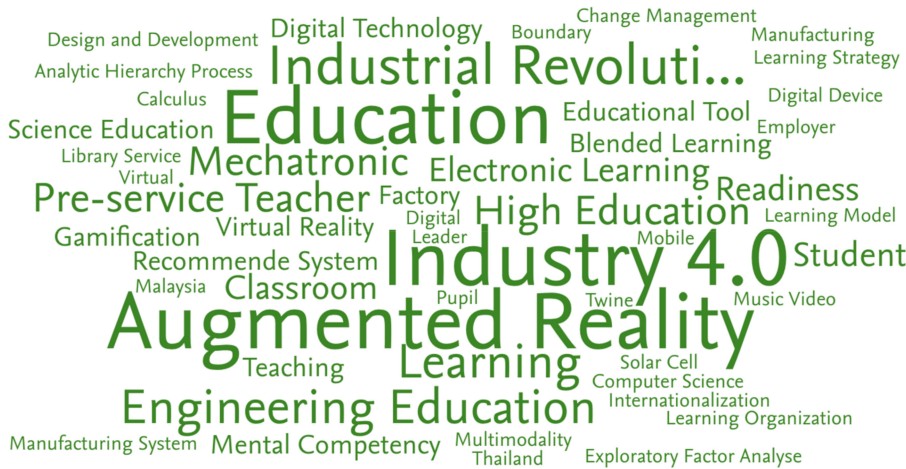

**Figure 2.** Top 50 key phrases by relevance, based on 87 publications about "Education 4.0" and "teaching" [10].

In this literature review of Education 4.0, publications on virtual reality, Industry 4.0, engineering education, and industrial revolution stood out, thus leaving the opportunity to contribute with studies focused on teachers. According to UNESCO [30], a teacher's profile is defined as the set of competencies that include abilities, aptitudes, skills, attitudes, and values that are put into practice in the classroom to teach students to build their knowledge and competencies to perform in the workplace. Therefore, the aim of this study was to identify the teaching profile required in the new undergraduate programs within the framework of Education 4.0 in order to contribute to decision-making for teacher recruitment, training and evaluation, talent management, and decisions for institutional policies interested in connecting competences with the needs of society. The theoretical support for the study was selected based on the advances that have been recorded in the teaching role, the characteristics of Education 4.0, the framing of the work that has been done (at the international, regional, and disciplinary levels), and the detection of the opportunity to contribute to teaching profiles for Education 4.0. The exploratory and descriptive approaches were supported by quantitative and qualitative instruments applied to undergraduate students, graduates, teachers, and experts in the educational field. The results are presented, and the data with triangulated sources and instruments are discussed [31]. The article ends with some conclusions that articulate crucial Education 4.0 teacher characterizations and delineate its limitations and contributions when considering future studies.

## 2. Materials and Methods

Currently, it is evident that continuous improvement processes in undergraduate programs responsible for the training of future teachers must be present in university agendas. Incorporating innovative approaches, teacher training, and cutting-edge online education are required elements when designing or redesigning innovative programs for the area of education. In the face of these requirements, the role of the teacher is fundamental, and there is currently is a gap between the current teacher profile and the required one.

The purpose of this study was to identify the teaching profile required in the new undergraduate programs within the framework of Education 4.0 in order to contribute to decision-making for teacher recruitment, training and evaluation, talent management, and decisions for institutional policies interested in connecting competences with the needs of society. The starting point was the question: What are the characteristics of the teaching role for Education 4.0? To answer this question, we used four instruments to collect data from students, graduates, teachers, and experts.

Based on the objective and the research question, we chose a descriptive and exploratory approach in which quantitative and qualitative instruments were applied; this approach was characterized by describing, analyzing, and interpreting complex phenomena from a social perspective [32]. The qualitative method was used to analyze data without numerical measurements by using descriptors, observations, attitudes, thoughts, and motivations [33]. For the collection of the information, we conducted (a) surveys, (b) interviews, and (c) focus groups.

## 2.1. Participants

The target population consisted of 337 undergraduate students in education programs and 313 graduates. We also included interviews with 20 experts in the educational field and five focus groups with 32 chancellors, school principals, university professors, and specialists in the educational area. We used databases provided by the university and secondary databases for this research. To define the sample, we applied a formula and weighting for finite universes.

## 2.2. Sample Calculation

Because the population was known, the finite universe formula was used to calculate the sample, with the following parameters: a 95% confidence level and a margin of error of 5%, where N is the total population, $Z\alpha$ is 1.96 squared (99% certainty), p is the expected proportion (in this case 50% = 0.5), q is 1 - p (in this case $1-0.5 = 0.5$), and e is (in this research) 5%.

After applying the formula, the sample size corresponded to 337 surveys focused on current students of the institution under study. It is important to clarify that for the segment of interest related to the graduates of Educational Sciences of the institution under study, the same formula described above was used for finite universes because the real population of interest was known. The total sample comprised 313 graduates.

### Geographical Distribution

In order to apply the surveys, the most representative cities in Ecuador were selected according to the number of inhabitants: Quito, Guayaquil, Cuenca, Loja, Machala, Portoviejo, and Azogues; in addition, it was analyzed that in these cities, the university under study has maintained a representative number of students. In the case of the graduates, a stratified sample was selected from each of the cities of Ecuador because there are professional graduates of the university in each of these cities (see Table 1), so it was considered important to know the perception of students and graduates on the received and required training in their profession in the face of current challenges.

## 2.3. Instruments

The surveys were taken from a Mexican university [34] and adapted to the Ecuadorian context. Before its application, the instrument underwent a validation process by experts. This process consisted of 31 structured questions that explored the units of analysis of the current teacher's profile, the profile of the student in education programs, the competencies required for the teacher, and the characteristics of an innovative teacher. Thus, the survey applied to students analyzed several topics: (a) motivation to study sciences and education programs, (b) current or future needs of the environment that should be present in the curriculum of the program, and (c) competencies developed and required in the profile of this profession. The survey was administered by using SurveyMonkey. The questionnaires were sent through e-mails and telephone calls.

The results were put in the Excel and SPSS (v.22.0) format for analyses. Additionally, there were 20 in-depth interviews conducted with experts in education from Mexico, Spain, Colombia, and Ecuador. The NVivo11 software was used for the data analysis of these interviews, and we conducted a content analysis of the participants' answers to determine the intentions of the participants within the context of the research.

**Table 1.** Distribution of students and graduates by Ecuadorian cities.

| City | Current Students [1] | Percentage % | Graduates [2] | Percentage % |
|---|---|---|---|---|
| Quito | 178 | 52.8 | 82 | 26.2 |
| Guayaquil | 37 | 10.9 | 31 | 9.9 |
| Cuenca | 49 | 14.6 | 37 | 11.8 |
| Loja | 39 | 11.6 | 44 | 14.1 |
| Machala | 20 | 6.1 | 27 | 8.6 |
| Portoviejo | 6 | 1.7 | 4 | 1.3 |
| Azogues | 8 | 2.3 | 24 | 17.7 |
| Cotopaxi | N/A | N/A | 10 | 3.2 |
| Ibarra | N/A | N/A | 10 | 2.9 |
| Chimborazo | N/A | N/A | 9 | 2.9 |
| Santo Domingo | N/A | N/A | 5 | 1.6 |
| Carchi | N/A | N/A | 5 | 1.6 |
| Esmeraldas | N/A | N/A | 4 | 1.3 |
| Bolívar | N/A | N/A | 3 | 1.0 |
| Zamora | N/A | N/A | 3 | 1.0 |
| Morona Santiago | N/A | N/A | 4 | 1.3 |
| Los Ríos | N/A | N/A | 2 | 0.6 |
| Orellana | N/A | N/A | 2 | 0.6 |
| Tungurahua | N/A | N/A | 2 | 0.6 |
| Sucumbios | N/A | N/A | 1 | 0.3 |
| Galápagos | N/A | N/A | 1 | 0.3 |
| Santa Elena | N/A | N/A | 2 | 0.6 |
| Otras | N/A | N/A | 2 | 0.6 |
| Total | 337 | 100 | 313 | 100 |

[1] Database of the population of current UTPL students (academic period from October 2019 to February 2020).
[2] Database of the population of UTPL graduates (2016–2019) provided by the graduate unit. N/A: not applied to the study

The design of the semi-structured interview, which was conducted with the experts in education, considered topics related to (a) main problems present in the educational system of Ecuador, (b) problems and weaknesses of the graduates of Educational Sciences, (c) technological/social/legal/ethical/political/environmental trends or megatrends that need to be incorporated into the training of professionals in the area of education, (d) ruptures or important changes that have occurred in recent years in the educational fields, and (e) profile that those professionals in charge of training in education should have.

The focus groups yielded units-of-analysis data on the current teacher profile, teachers' competencies, and innovative teachers' characteristics. There were four focus group sessions with 32 people, including educators, chancellors, school principals, university professors, and specialists in the educational area. The NVivo11 software was used to analyze the information, and we again conducted a content analysis of the participants' answers to determine the perspective of the participants within the context of the research study [35].

In the four focus groups developed through virtual connection, three specific topics were analyzed with the participants: (a) technological, economic, political, environmental and social changes that will have an impact on training and the education labor market in the coming years; (b) disruptions or changes that will have an impact on training and the education labor market in the coming years; and (c) the profile that professionals in charge

of training in education should have. Each of the focus groups lasted approximately two hours, with an average attendance of 7–10 participants.

### 2.4. Data Analysis

The application of quantitative and qualitative instruments made it possible to have greater validity in the results. The triangulation analysis of the data came from three actors: students, graduates, and expert teachers and managers in education. This analysis made it possible to clarify and complement the collected information while considering the different stakeholders' perspectives [33]. Ethical aspects were taken care of in the collected data through permissions to use the information for academic purposes. The data management was objectively handled to provide valid evidence. Finally, the privacy of the participants' data adhered to the institutional policies involved in the study.

## 3. Results

The results derived from the units-of-analysis data are presented below.

### 3.1. Current Teacher Profile

To determine the teacher profile, we conducted focus groups and interviews. The characteristics by gender and occupation of the participants are shown in Table 2.

**Table 2.** Focus group participants and interviewees characteristics (own elaboration).

| Variable | Focus Group (N = 32) | | Interviewees (N = 20) | |
|---|---|---|---|---|
| | N | % | N | % |
| Gender | | | | |
| Female | 18 | 56.3 | 4 | 20 |
| Male | 14 | 43.7 | 16 | 80 |
| Total | 32 | 100 | 20 | 100 |
| Occupation | | | | |
| Chancellor/Vice-Chancellor | 3 | 9.4 | 2 | 10 |
| School Principal | 11 | 34.3 | 3 | 15 |
| Professor | 10 | 31.2 | 7 | 35 |
| Specialist | 8 | 25.1 | 8 | 40 |
| Total | 32 | 100 | 20 | 100 |

According to the people consulted in the focus groups and interviews, the existing teacher profile was found to not meet current contextual needs. In the sense that the teacher's profile should lead to achieving educational objectives and future generations' training, the people who participated in the focus groups determined a series of teacher training deficiencies. From the transcripts of the different sessions that took place, the following reflections emerged.

The classes were still lecture-based, so an urgent change was required in the pedagogical processes carried out for education programs. For example, a comment in one of the groups was, "The teacher needs to master constructivism. The universities are familiar with this methodology, but a little more is needed for development. Time and experience mold the training, and the teachers must investigate to ensure that their students get the skills they need".

Another of detected problem was the lack of knowledge in the use of technology. Thus, a group commented that "the pandemic revealed how unprepared the education programs were using new technologies for teaching purposes." Another group mentioned that "both students and teachers are unfamiliar with tools that facilitate learning in completely virtual environments".

The participants felt there is a lack of appreciation for teachers' work and that those who enter this profession do not do so out of vocation but because of a lack of job opportunities in other areas. One of the groups indicated that "the teacher feels undervalued, burdened with administrative work, pressured by peers, with a low social valuation of the work".

Additionally, they mentioned the absence of creative or divergent thinking and the presence of reading problems. They commented that "The teachers have low reading levels, i.e., an initial reading comprehension level," and "very little ability for divergent thinking, understanding education as transmitting knowledge, and not . . . sharing and spreading what they are passionate about." Finally, there was a problem with the teaching practice, as commented on in one of the groups: "Direct interrelationships are required in the field. The practices and the monitoring of practices by the authorities of the institution where they are carried out should be improved".

These reflections also agreed with the experts' responses to the interviews. They mentioned that "education programs are perceived as outdated and focused on the repetition of knowledge and memorization." In addition, there was found to be little connection between theory and practice, nor was there a connection to the realities experienced by teachers.

The experts also commented that due to the pandemic, the lack of preparation of teachers and students to work in virtual environments was evident, and "Programs in education are perceived as rigid and based on the 'should be;' so curricular adaptations cannot be made".

### 3.2. Student Profile in Education Programs

To determine the current student profile in education programs, we analyzed the survey results for current students at the Ecuadorian university (object of study) and the survey for graduates of the same university. These results showed that current students' age in education programs was 19–40 years, with the highest percentage between 19 and 25. The graduates were between 24 and 45, with the highest occurrence between 26 and 40. In both groups, the majority were women, and most of them had graduated or were studying Primary Education, as shown in Table 3.

Almost half of the current students were working in the public or private sectors. In the case of graduates, the majority worked in private companies and educational institutions. The salary difference between students and those who had completed their studies was an average of 60% more for the latter. In this regard, 178 students (52.8%) mentioned having a salary of less than $400, while the graduates who were working (242)—which corresponded to the majority (57%)—stated having an income between $767 and $1106.

The characteristics of the programs most sought by students were teaching quality, certifications, scholarships, modality, price, and offering of undergraduate programs, as shown in Figure 3. Additionally, the consulted former students mentioned flexibility, technology, and administrative attention as attractive elements for choosing an academic discipline.

In terms of teaching modalities, current students were found to require that the virtual modality have adequate synchronous sessions and be flexible to facilitate their work and personal life, as well as to allow them to sufficiently interact with teachers and classmates. For education graduates, their interest in continuing with a program would be in the distance mode because it enables them to continue their work and personal life. The characteristics they were found to desire were adequate synchronous sessions, excellent teaching and learning strategies, and sufficient interaction with teachers and classmates, as shown in Figure 4. The differences between students and graduates allowed us to know which perspectives fit the modality to continue studying a career in education.

The choice stated by the graduates to study a program in education was based on their vocation, family opinions, and the earning potential of their profession. Additionally, respondents believed that education programs should include digital transformation (80.2%), reasoning for complexity (78.3%), ethical and citizen engagement (78%), social intelligence

(75.1%), self-management skills (73.5%), inclusion and respect for diversity (70.9%), social responsibility (65.5%), and innovative entrepreneurship (45.4%) (see Figure 5).

**Table 3.** Participants' characteristics (own elaboration).

| Variable | Current Students (N = 337) | | Graduates (N = 313) | |
|---|---|---|---|---|
| | N | % | N | % |
| Age | | | | |
| 19–25 | 126 | 37.4 | 9 | 3 |
| 26–33 | 108 | 32 | 112 | 36 |
| 34–40 | 81 | 24 | 107 | 34 |
| 41–48 | 18 | 5.4 | 49 | 16 |
| 49–55 | 4 | 1.2 | 24 | 7.7 |
| 56–66 | 0 | 0 | 12 | 3.3 |
| Total | 337 | 100 | 313 | 100 |
| Gender | | | | |
| Female | 258 | 76.6 | 68 | 21.7 |
| Male | 79 | 23.4 | 245 | 78.3 |
| Total | 337 | 100 | 313 | 100 |
| Program they study | | | | |
| Basic Education | 98 | 29.1 | 106 | 33.9 |
| Primary Education | 92 | 27.3 | 62 | 19.8 |
| Science pedagogy | 55 | 16.3 | 0 | 0 |
| Language Pedagogy and Literature | 75 | 22.2 | 20 | 6.4 |
| Pedagogy of Experimental Sciences | 9 | 2.7 | 0 | 0 |
| Human Sciences and Religious | 8 | 2.4 | 0 | 0 |
| ES/Chemistry, Biology, Physics, Math. | 0 | 0 | 46 | 14.7 |
| ES/English | 0 | 0 | 79 | 25.2 |
| Total | 337 | 100 | 313 | 100 |
| Occupation | | | | |
| Students | 178 | 52.8 | 5 | 1.6 |
| Workers | 113 | 33.5 | 224 | 71.6 |
| Unemployed | 40 | 11.9 | 66 | 21 |
| They work and study | 0 | 0 | 18 | 5.8 |
| Other activities | 6 | 1.8 | 0 | 0 |
| Total | 337 | 100 | 313 | 100 |

According to the graduates' responses, at the end of their degree, 250 people indicated that they would like to study for a master's degree, 49 people would like to study for another degree, and 43 would like to study a specialty. In education, they mentioned an interest in continuing with primary education, university management, and educational management.

### 3.3. Competencies Required of the Teacher

According to the interviewees, the ideal teacher profile for Education 4.0 should maintain a balance between soft skills—such as leadership, motivation, and communication—and technological competencies and critical thinking. As one of the interviewees mentioned,

"The teacher must have a strong leadership profile and be focused on the students' needs. They must exert pedagogical leadership, being a leader who motivates and generates concerns, have mastery of ICT with an overall view of the global culture, and be involved in the globality of thought". Another interviewee commented, "In the technological field, teachers require the necessary competencies for digital pedagogy".

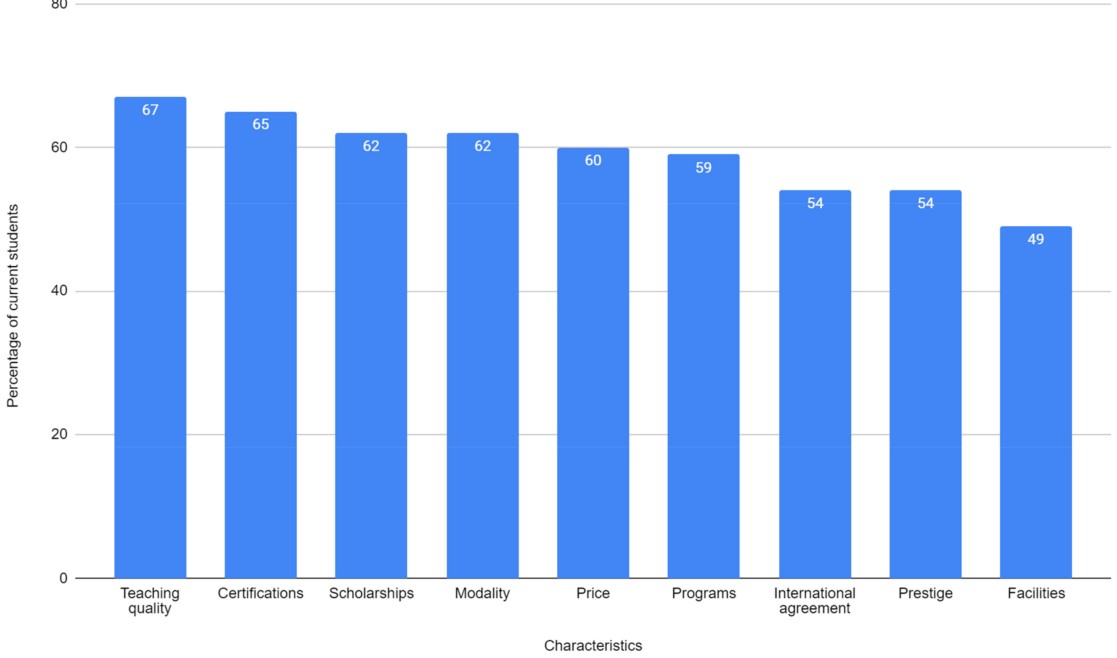

**Figure 3.** Characteristics of the most requested academic programs by current students (own elaboration).

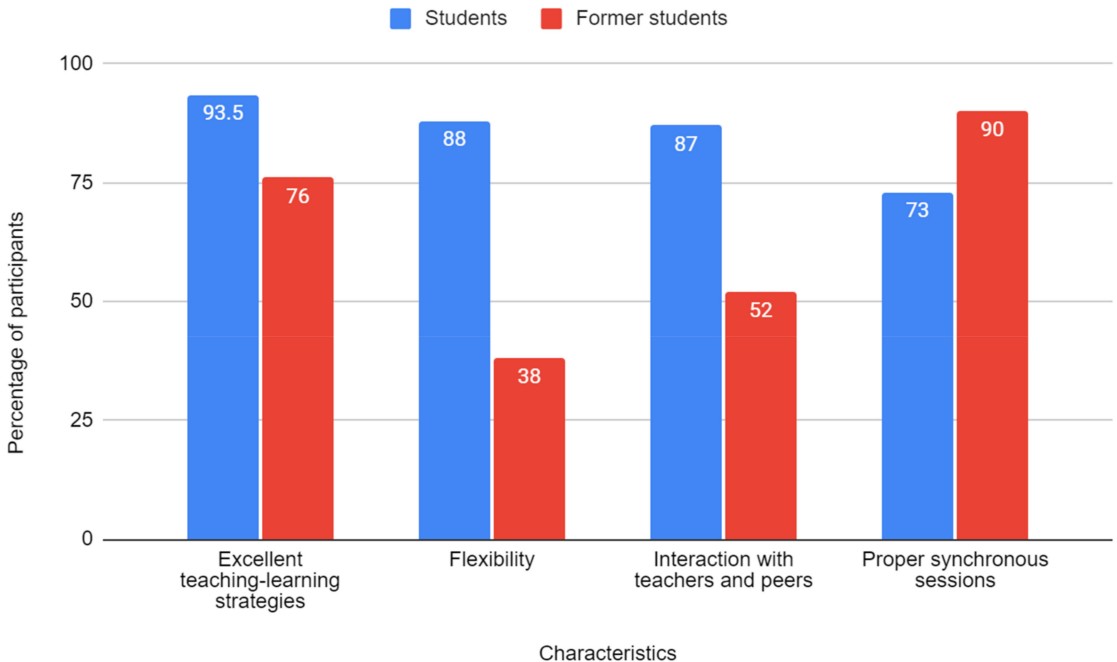

**Figure 4.** Characteristics of the online programs (own elaboration).

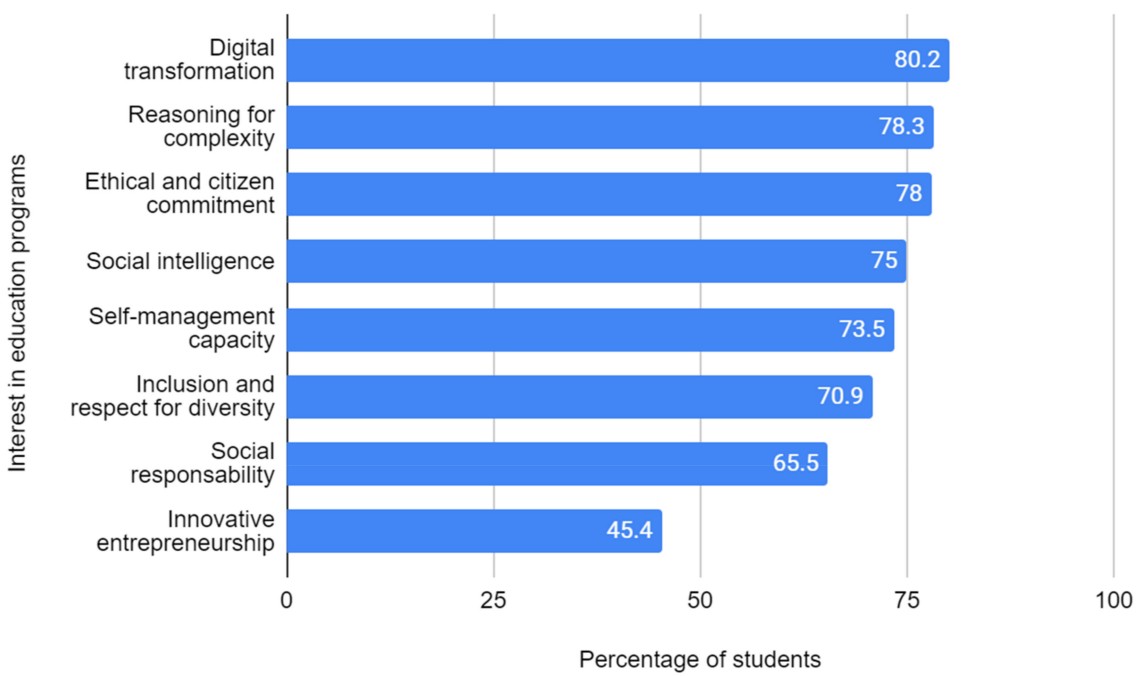

**Figure 5.** Student interests in education programs (own elaboration).

Regarding soft competencies, such as communication, it was mentioned that "Educators should be experts in communication, in written elements, and interpersonal communications [and] competencies of the teacher of the 21st century". They also mentioned having competencies to work not only by objectives but also by carrying out formative actions: "How do we want them to apply this knowledge? How do we want them to put it in a real practical context?" These questions lead teachers to curricular designs that counteract a repetitive education that is based on memorization and oriented more to practice and problem-solving.

The teacher of the future should be critical and have a greater capacity for analysis. As one of the experts mentioned that "The profile should be that of a research teacher, one able to diagnose the problems of his or her context and propose various solution projects". It also requires a much more human, empathetic teacher who becomes a facilitator and a coach for students, as mentioned by the experts: "The teacher must have didactic competencies and humanistic, relational competencies"; "the teacher must be a guide, a facilitator, the one who supports, who guides, who follows, who leads, and who facilitates their students' learning"; and "assumes the role of a coach and designer of learning experiences—not a repeater, nor a transmitter".

*3.4. Characteristics of an Innovative Teacher*

According to the experts, the profile of the teacher must be adjusted to meet the new requirements. In this sense, universities must train professionals in education with different competencies and skills and prepare them for a role that differs from the traditional teacher. There must be a balance between soft skills and technological competencies. For example, one of the experts commented that "the graduate in education must have a good handle on managing information through technologies" and "training new professionals who understand the society we live in".

Among the competencies mentioned were problem-solving, developing playful and practical activities, creativity, research, accompaniment and training, and skills with distance learning devices. Additionally mentioned were critical and creative thinking, research, innovative entrepreneurship, knowledge of the subject, organization, human sense,

empathy, ability to evaluate, self-management, ethical and citizen commitment, digital transformation, social intelligence, and innovation.

Another comment from the experts was that "A deeply humanistic person is needed, one who has an elaborated and developed critical thinking and is aware of the main problems of society". Another expert mentioned that "The teacher must be a person who has a collaborative spirit, who can share his knowledge, who can unlearn to learn again, and who can work on a team with teachers and managers to transfer this knowledge to his students and be the example to which the students aspire". The teacher "not only imparts knowledge but also generates or ignites their interest; the teacher is more of a facilitator of learning".

The teacher's role is no longer one who only shares knowledge. They are facilitators who promote reflection and bring that knowledge closer to the students' reality. The experts commented: "So, the teacher of the 21st century is a facilitator, a collaborator, a digitally competent person, one who works in a team and learns continuously. The last one is crucial. The teacher must be a person who is continuously researching, learning, and applying and experimenting in their classes".

The teacher must also master other languages and know new technologies like artificial intelligence and digital technologies with educational applications. They must promote critical thinking, design study plans that generate learning for the students' reality, and develop projects that impact different community sectors. The characteristics of the teacher profile are presented in Figure 6.

**Figure 6.** Teacher's profile to meet the new requirements (own elaboration).

## 4. Discussion

Teachers play a crucial role in current education, especially for the future professionals of Educational Sciences, because they will be the next educators. Teachers are responsible for the transformation of the students' learning model. According to the results, the profile of the current teacher has characteristics of a traditional teaching model. Hernandez et al. [26] stated that current students of education have not integrated the theoretical and practical knowledge required to perform their roles at the end of their training. Many cannot design and implement programs for pedagogical intervention with a student-centered teaching approach in which profound learning is developed. The interviewed experts agreed with Alsina [27] that universities must guarantee that future teachers develop self-regulation during their training process, have elements to mark their behavior, assume goals and design strategies to achieve them, and critically and objectively self-evaluate their results. The teaching profile is postulated as a substantial element for the formation of students in these new scenarios.

Education 4.0 students apply for training programs and accreditations that certify competencies, supported by consolidated academic, technological, and administrative aspects. Figure 3 shows the characteristics of the programs most sought after by students, where

teaching quality, certifications, scholarships, flexibility, technological aspects, and administrative support are valued as attractive elements for choosing a program. Miranda et al. [28] agreed with the pillars of Education 4.0, such as training by competencies, active learning methods, information and communication technologies, and infrastructure, all components necessary for designing new educational innovation projects. The curricular design faces the challenge of consolidating students' expectations with the university support available to teachers.

Designing flexible environments to solve real problems is one of the challenges of strengthening Education 4.0. Figure 4 shows that both current education students and graduates require the virtual modality to be flexible to facilitate the continuation of their work and personal lives and to allow them to sufficiently interact with teachers and classmates through adequate synchronous sessions. Additionally, the lack of valuing the teaching program is a result that concerned the population in the analyzed sample. Making this profession attractive and valuing teachers' responsibility and social commitment are pending tasks for governments and universities. Accomplishing this should improve the selection choices of those entering this program [28]. Linkage to the real world and course flexibility are elements to consider when designing education curricula that accommodate digital transformation.

This digital transformation requires a commitment to change and innovation from academic actors in Education 4.0 environments. As Figure 5 shows, the student profile of the teaching discipline must integrate competencies related to digital transformation, reasoning for complexity, ethical and citizen commitment, social intelligence, capacity for self-management, inclusion and respect for diversity, social responsibility, and innovative entrepreneurship. These competencies are also required in an Education 4.0 model [28], which incorporates innovative learning methods, didactic, pedagogical, social, and technological competencies, and others. The transversal training to develop digital skills in students should be an essential point in training programs.

The teacher profile in Education 4.0 requires disciplinary and transversal competencies, where digital competencies are also a constant, as with students. According to this investigation's results, the ideal teacher profile for Education 4.0 should have both technological competencies and soft skills. Other authors, such as Carvalho et al. [3], coincide; they indicate the need to develop digital literacy in teachers to favor educational quality. This is because implementing technology-based educational experiences can positively impact student outcomes [5], provided that it is accompanied by teacher training that allows the development of other skills such as motivation, self-efficacy, responsibility, and the ability to innovate [16]. These are competencies required for the teacher to assume a role as a facilitator of student learning where the student participates actively in their training.

The competencies required in a teacher for the new scenarios are related to innovation, problem-solving, creativity, critical thinking, research, and entrepreneurship. The results in Figure 3 show what is desirable in the profile of teachers. The teacher must promote these in his/her students, as mentioned by Peredrienko et al. [17]. These attitudes will be of great value for the student to integrate into the labor market in a context that requires knowledge for an increasingly digitalized world. In the consultations with the experts, we determined that education must stop being traditional and relying on memorization, elements that stifle the development of creativity and innovation; it must favor entrepreneurship and research. In this sense, institutions should develop pedagogies that encourage these processes and allow teachers to guide students towards creating experiences facilitated by technologies [24]. Hence, a teacher leadership profile is proposed to promote digital pedagogies in the context of the current reality.

The knowledge that the future teacher must have about their role is a determining element to achieve success in students' education. In turn, universities must train professionals prepared for Education 4.0. This study recognizes that teachers must have a human feeling, ethical commitment, and social intelligence and have the necessary technological and soft skills. This finding is supported by research by Serdyukov [16], where the teacher

must integrate the different environments of education and society, creating a suitable atmosphere to develop competencies in the student. Additionally, as supported by other research, intelligent learning environments that allow students to adapt to changes, offer opportunities for reflection, and be attentive to their sensitivity should be fostered [22]. It is considered that these characteristics of the teacher will allow the generation of spaces that enhance innovative ideas, are sensitive to the environment and take advantage of technologies to benefit citizens' training.

## 5. Conclusions

It is indisputable that society is changing rapidly, especially in the last year due to the health contingency caused by COVID19. In this context, the need arises for a professional in education who can lead these processes of change to train citizens who are better prepared for the 21st century. Hence, the results of our research show a teaching profile with characteristics that can guide the processes for Education 4.0. The students, teachers and experts who participated in this study were able to recognize elements that characterize the Education 4.0 teacher such as (a) designs strategies for competency-based training through active learning methods; (b) has soft skills, such as digital transformation competencies, reasoning for complexity, ethical and citizen commitment, social intelligence, capacity for self-management, inclusion and respect for diversity, social responsibility, and innovative entrepreneurship; (c) has human sensitivity and trains students to develop ethical behavior and social intelligence, integrating the educational environment and society; and (d) uses technologies and applies new tools that facilitate learning through virtuality, artificial intelligence, digital technologies, and educational applications.

According to this research, the profile of the Education 4.0 teacher can have great significance in different areas. The results may be valuable to administrators, educational and social entrepreneurs, and trainers because they shape the teaching profile as a specialized professional with competencies for innovation, complex problem solving, entrepreneurship, collaboration, international perspective, leadership, and connection with the needs of society. For policy-makers, the characteristics of this teaching profile can guide training programs that strengthen the teachers' labor, promote quality education, and broaden the vision of educational institutions. For universities, this professional can contribute to innovation, propose improvement strategies, support decision-making and contribute to problem-solving in research, as shown in Figure 7.

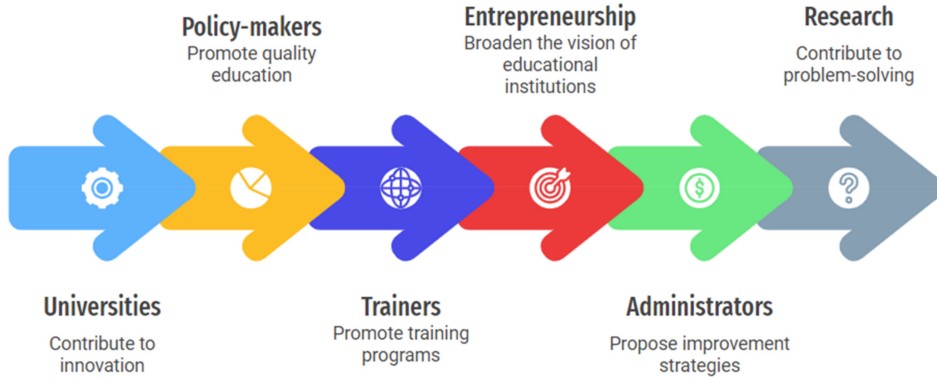

**Figure 7.** Significance of the Education 4.0 Teacher (own elaboration).

The presented study was based on an exploratory and descriptive analysis where students, graduates, professors, experts, and decision-makers (chancellors, school principals, university professors, and specialists in the educational field) were consulted to answer the question: What are the characteristics of a teacher in the context of Education 4.0? Though our objective was not to generalize, we believe that the results of this study can be implemented in educational services that need to update their teacher training curriculum.



The characteristics of the Education 4.0 teacher identified in the study can guide training processes to better prepare students for life and work.

The limitations of the research included that the obtained results were limited to a small study group that did not represent the general population. However, the description of the Education 4.0 teacher profile can be used as a guide for further research on this topic, which has been little explored in the literature. Future research can be oriented towards strengthening the teaching profile for Education 4.0 by taking the findings of this research in university curricula to explore ways to integrate other educational community members, industry, and social actors to develop innovation competencies. Though this study focused on teachers' profile for programs related to education, the findings may be of value for the characterization of teachers in other disciplines, in the sense that they are also trainers of talent.

**Author Contributions:** Conceptualization, M.S.R.-M. and M.I.L.-A.; methodology, M.P.-C.; software, M.P.-C.; validation, A.Z.-O. and M.P.-C.; formal analysis, M.I.L.-A., A.Z.-O. and M.P.-C.; investigation, M.S.R.-M. and M.I.L.-A.; resources, M.I.L.-A.; data curation, M.S.R.-M.; writing-original draft preparation, M.P.-C.; writing-review and editing, M.S.R.-M.; visualization, M.I.L.-A.; supervision, M.S.R.-M. and M.I.L.-A.; project administration, A.Z.-O.; funding acquisition, M.I.L.-A. All authors have read and agreed to the published version of the manuscript.

**Funding:** This study was supported by the Rector's Office of the Universidad Técnica Particular de Loja (Ecuador) through its Vice Rector's Offices for Innovation and Distance Education. The project was developed in collaboration with Tecnologico de Monterrey (Mexico).

**Data Availability Statement:** Not Applicable, the study does not report any data.

**Acknowledgments:** This paper is a product of the project "OpenSocialLab: linking experiential learning to scale levels of mastery in social entrepreneurship skills," with funding from the NOVUS 2019 Fund. The support of Tecnologico de Monterrey for educational innovation projects is appreciated (Agreement: Novus 2019). The authors would like to acknowledge the technical support of Writing Lab, Institute for the Future of Learning, Tecnologico de Monterrey, Mexico, in this work.

**Conflicts of Interest:** The authors declare no conflict of interest.

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
