# Peer review of "Characterization of the Teaching Profile within the Framework of Education 4.0"

_futureinternet, doi:10.3390/fi13040091_

Round 1

Reviewer 1 Report

This paper presents very timely topic.

A few words about previous generations of Education 1.0 to Education 3.0. would be worth to recall. And what is the “delta” of Education 4.0 from these previous generations, what is the difference of this desired approach to learning from past experience. Since the article was offered in “Future internet” journal, it may be a good idea to highlight the connection between Education4.0 and the future Internet.

The workflow of the study must be described. What is the „formula“ and how „weighting for finite universes“ was determined [“we applied a formula and weighting for finite universes” – page 4].

The results section seems scattered, a better structure for presenting the results and a more in-depth analysis would be useful.

Table 1 needs to be revised. The title of the first line indicates percentages, which, however, are missing as data.

Figure 3 contains an incomplete legend - 8 are the sectors, 7 are indicated. Furthermore, it is not clear these "student interests" in what are? The description of the figure is not complete.

Some supplementary materials used for the investigation would be suitable to be provided.

Author Response

Dear Reviewer 1, Thanks for the review. We attach the updated version with our responses.

Kind regards,

Reviewer 2 Report

Please refer to the appended file.

Author Response

Dear Reviewer 2,

Thanks for the review. We attach the updated version with our responses.

Kind regards,

Reviewer 3 Report

The contribution is quite original, but still interesting for the magazine's readers. The presentation of the research is clear and scientifically sound; the objectives, methods and discussion of the results are clear. Therefore, the contribution is worthy of being published without making changes.

Author Response

Dear Reviewer 3,

Thanks for the review. We attach the updated version with our responses.

Kind regards,

Round 2

Reviewer 1 Report

Most of my remarks were answered. Thanks!

Author Response

Attention to Evaluator's Review 1

Dear evaluator, thank you very much for the review of the article. We appreciate the time you spent for the evaluation.

Kind regards,

Reviewer 2 Report

Re-review

The authors did an excellent job of improving their text.

It may be published in the current version, or authors may improve some minor issues.

  1. I appreciate the work Authors provided to improve the previous version of the text.
  2. The goal formulation.
    • Authors persist: the purpose was to identify
    • The authors explain why they identify.
    • I accept goal formulation
  3. I accept the merit content of the new version of the Introduction.
  4. There are still some issues that may be improved.
    • Figure 1. There is no sense to include it. It is just a copy of the web service. There is no information to which date it refers. It is unreadable, e.g. all data for Europe are on one spot, covering one another. Maybe the table completed with some verbal analysis would have some sense.
    • Figure 2. There is no sense to include it. It is just a copy from the web service. There is no information to which date it referes to. It is unreadable, e.g. there is a lot of yellow bullets with egual size. What do they represent? Spots are covering each other. Maybe the table completed with some verbal analysis would have some sense.
    • Figure 3. There is no information to which date it refers to. Maybe the table completed with some verbal analysis would be of more use.
    • The comment: “(*) 50% is used because it maximizes the sample size, since the proportion is not known”. It is a highly unprecise statement. It should be elaborated. The probability density assumption should be given, and the sample drawing scheme shpold be named, etc.
    • Table 1. The legend is entirely incomprehensible; it is not known which numbers refer to each database. Looks awfully. It should be redesigned.
    • Table 2. There are 20 respondents. Counting and showing percentages (%) in a situation where there are less than one hundred items doesn’t make sense. Looks awfully. It should be redesigned.
    • Table 3. Looks awfully. It should be redesigned.
    • Figure 4. It is not clear who was the respondents. All; current students; graduates?
    • Figure 5. The reasons for the differentiation between students and graduates should be touched upon.

Author Response

Dear evaluator, thank you very much for the new comments to the article. We we attach the version with the changes required.

Best regards,
